# Effectiveness of combining prevention psychological interventions with interventions that address the social determinants of mental health in low and middle-income countries: protocol of a systematic review and meta-analysis

Eleonora Prina [1], Beatrice Bano [1,2], Rakesh Singh [3,4], Emiliano Albanese,[2,5] Daniela Trujillo,[6] María Cecilia Dedios Sanguineti,[7] Katherine Sorsdahl,[8] Nagendra P Luitel [3], Emily C Garman,[8] Marianna Purgato [1,9] Corrado Barbui,[1,9] Mark J D Jordans [4], Crick Lund [4,8]

For numbered affiliations see end of article.

**Correspondence to**
Dr Eleonora Prina;
eleonora.prina@univr.it

## ABSTRACT

**Introduction** Common mental health conditions (CMHCs), including depression, anxiety and post-traumatic stress disorder (PTSD), are highly prevalent in low and middle-income countries (LMICs). Preventive strategies combining psychological interventions with interventions addressing the social determinants of mental health may represent a key strategy for effectively preventing CMHCs. However, no systematic reviews have evaluated the effectiveness of these combined intervention strategies for preventing CMHCs.

**Methods and analysis** This systematic review will include randomised controlled trials (RCTs) focused on the effectiveness of interventions that combine preventive psychological interventions with interventions that address the social determinants of mental health in LMICs. Primary outcome is the frequency of depression, anxiety or PTSD at postintervention as determined by a formal diagnostic tool or any other standardised criteria. We will search Epistemonikos, Cochrane Controlled Trials Register (CENTRAL), MEDLINE, Embase, PsycINFO, CINAHL, Global Index Medicus, ClinicalTrials.gov (Ctgov), International Clinical Trials Registry Platform (ICTRP). Two reviewers will independently extract the data and evaluate the risk of bias of included studies using the Cochrane risk of bias tool 2. Random-effects meta-analyses will be performed, and certainty of evidence will be rated using the Grading of Recommendations Assessment, Development and Evaluation approach.

**Ethics and dissemination** This study uses data from published studies; therefore, ethical review is not required. Findings will be presented in a published manuscript.

**Trial registration number** CRD42023451072

## INTRODUCTION

Common mental health conditions (CMHCs), defined here as depression, anxiety and post-traumatic stress disorder (PTSD), are highly prevalent in low and middle-income

## STRENGTHS AND LIMITATIONS OF THIS STUDY

⇒ This study aims to address an important scientific gap related to the lack of evidence-based interventions combining psychological and social ingredients for preventing common mental health conditions in low and middle-income countries.

⇒ The present review will be conducted following the Cochrane Handbook for Systematic Reviews of Interventions and the Preferred Reporting Items for Systematic reviews and Meta-analyses 2020 guidelines for data reporting.

⇒ We expect high degrees of clinical and methodological heterogeneity among the included studies, as they have been conducted across different types of settings (including humanitarian settings).

countries (LMICs) and exert a significant contribution to the global burden of disease and disability.[1–4] The WHO estimates that the prevalence of CMHCs in conflict settings is at 22%,[5] which is notably, around five times higher than the general population.

In LMICs, a range of factors such as poverty, conflict exposure, malnutrition, gender-based violence and social inequalities influence the prevalence, incidence and prevention strategies of CMHCs.[6–10] Described as social determinants of mental health, these factors are the social and economic conditions that may have a direct influence on the prevalence and severity of mental disorders across the life course.[11–13] Evidence suggests that CMHCs are strongly socially determined among populations worldwide.[14]

The COVID-19 pandemic has emerged as an additional public health crisis within an already complex setting of psychological distress.[15–17] The pandemic has not only exacerbated existing psychological challenges but has also significantly affected the social determinants of mental health.[18]

In LMICs, the gap between persons in need of mental health interventions and those accessing such services is large, with a high burden associated with CMHCs.[19] Reports from the WHO highlight persistent disparities in mental health resource availability, significant differences between high-income and low-income countries as well as across regions. They emphasise the need for sustained national-level investment in mental healthcare policies, plans, health services and monitoring systems.[20] Therefore, the global mental health community advocates for the implementation of prevention strategies as an important opportunity for addressing this mental health gap.[21]

Prevention is an approach aimed at reducing the likelihood of future disorders in the general population or for people who are identified as being at risk of a disorder.[22 23] Psychological prevention interventions in LMICs includes brief, straightforward interventions, mostly delivered via a task-shifting approach defined by the WHO as *the rational redistribution of tasks among health workforce teams*.[24] Specific functions are shifted and reallocated, where appropriate, from highly qualified health professionals (such as psychiatric, psychotherapists) to health workers with shorter training and fewer qualifications.

Preventive strategies can range from universal, selective and/or indicated interventions according to the Institute of Medicine (IOM) Framework.[22 25] Specifically, universal prevention includes strategies for the whole population, independently from the exposure to risk factors. Selective prevention targets high risk subpopulations based on identifiable risk factors[26] and indicated prevention focuses on individuals with increased vulnerability for a disorder (ie, increased psychological symptoms) but not meeting the criteria for a full-blown mental health condition.[22 25] Psychological interventions are widely used in the prevention of CMHCs, as highlighted by the Mental Health Gap Action Program Intervention Guide.[27] These interventions offer practical support, aimed at enhancing coping strategies, hope and resilience. Two recent Cochrane reviews focused on the effectiveness of prevention interventions. A systematic review of 113 randomised controlled trials (RCTs) with 32 992 participants across 39 LMICs identified a positive effect of prevention interventions on symptoms of depression (SMD −0.69, 95% CI −1.08 to −0.30), anxiety (MD −0.14, 95% CI −0.27 to −0.01) and PTSD (SMD −0.24, 95% CI −0.41 to −0.08) for adult populations.[28] Accordingly, a Cochrane review focused on prevention interventions in humanitarian settings, found that psychological counselling may be effective for reducing depressive symptoms (MD −7.50, 95% CI −9.19 to −5.81) and anxiety symptoms (MD −6.10, 95% CI − 7.57 to −4.63) among individuals without a formal mental health diagnosis.[29] However, none of the included RCTs provided data on the efficacy of interventions to prevent the onset of CMHCs (ie, incidence).[30]

A recent Lancet Commission sought to align global mental health efforts with sustainable development goals and emphasised the importance of efforts to prevent mental health disorders.[21] While there is preliminary evidence supporting the effectiveness of psychological interventions[28] and programmes targeting the social determinants of mental health[31] in preventing mental health disorders, there is a notable absence of systematic reviews of trials combining social (ie, seeking to change or improve a social determinant of mental health) and psychological components. Only through our review we can understand the efficacy of both ingredients (ie, psychological, and social) and approach a 'component' meta-analysis.

This systematic review and meta-analysis aims to comprehensively evaluate the impact of interventions that include psychological ingredients and target the social determinants of mental health.

## METHOD AND ANALYSIS
The project started on 1 December 2023, and it is expected to be completed by 31 July 2024. This protocol is reported in line with Preferred Reporting Items for Systematic Reviews and Meta-Analyses (PRISMA-P) guidelines[32 33] (online supplemental appendix 1).

### Criteria for considering studies for this review
The inclusion criteria are reported below and visually represented in figure 1.

#### Type of studies
This systematic review and meta-analysis will include RCTs. Both individual and cluster RCTs will be considered. We will include trials that employ a cross-over design, and we will use data from the first randomised stage only.

#### Type of participants and settings
We will include participants of any age, gender, ethnicity and religion. Given the focus on prevention of CMHCs, we will exclude participants scoring above a validated cut-off on a symptom checklist at baseline or with a formal diagnosis of mental disorders at the time of recruitment, regardless of whether they have a physical health disability. We will focus on (1) the general population (universal prevention), (2) subpopulations exposed to risk factors, for example, populations exposed to armed conflicts or natural disasters, those living in poverty or without social networks (selective prevention), and (3) individuals with increased psychological symptoms, but not meeting the criteria for a diagnosed mental disorder (indicated prevention).

We will include studies recruiting participants in any setting (eg, healthcare setting, refugee camps, schools,

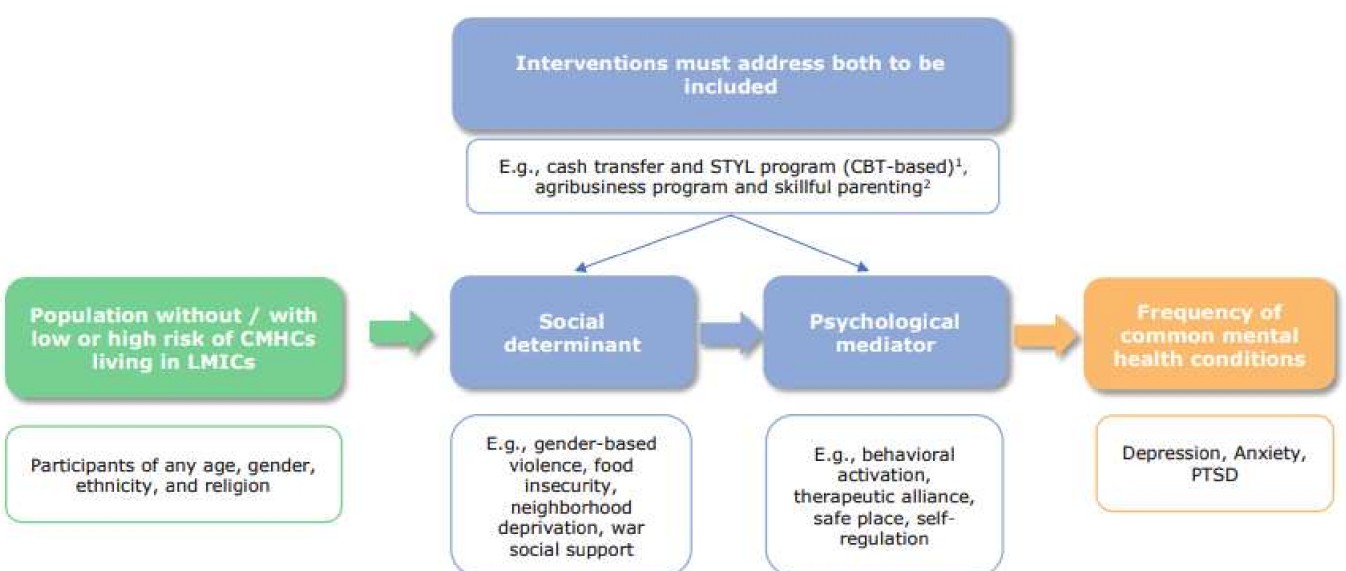

**Figure 1** Causal pathway indicating inclusion criteria for the review. CBT, cognitive behavior therapy; CMHCs, common mental health conditions; LMICs, low and middle-income countries; PTSD, post-traumatic stress disorder; STYL, Sustainable Transformation of Youth in Liberia. [1]Blattman *et al*[46] 2017. [2]Lachman *et al*[36] 2020.

home, community setting) in LMICs, as defined by the World Bank classification at the time of study conduction.[34]

## Type of interventions
We will include RCTs that combine psychological interventions with interventions that address the social determinants of mental health in LMICs. We will consider (1) interventions where both the social determinants and the psychological components are integrated into a single intervention (eg, Strengthening Evidence base on scHool-based intErventions for pRomoting adolescent health (SEHER) programme conducted in India,[35] which involve problem-solving counselling for health, social and academic issues, alongside school-wide strategies addressing themes like bullying, reproductive and sexual health, gender and violence, rights and responsibilities), (2) interventions in which the social and psychological components are delivered separately (eg, in a trial in Tanzania,[36] two programmes were evaluated: the Skillful Programme, a group-based intervention focusing on parenting skills and child protection, and the Agribusiness Training Program, which included three workshops covering access to credit for farm inputs, guidance on enhancing farming techniques and establishing market connections).

The social determinants of mental health will be categorised into five primary domains, according to the theoretical framework of Lund *et al*[11]: demographic, economic, neighbourhood, environmental events and social/cultural. Within each domain, these determinants may exert influence on individuals' mental health throughout their lives, involving a combination of distal and proximal factors.[11]

We will only include interventions delivered through task-shifting, for example, by non-specialist health workers (PHWs) and/or community workers (CWs) (ie,

interventions delivered by providers without a formal training in mental health and/or specialisation with a redistribution or delegation of healthcare tasks within workforces and communities) due to scalability and cost-effectiveness reasons.[21 37 38] Whenever possible, we will classify prevention interventions following the criteria of the IOM[25] as universal, selective and indicated.[22]

Interventions will be compared versus any control conditions such as no treatment, usual care, active control or waiting list.

## Type of outcome measures
Our primary outcome is the frequency of depression, anxiety or PTSD at postintervention, as determined by a formal diagnostic tool (ie, the Diagnostic and Statistical Manual of Mental Disorders (DSM),[39] International Classification of Diseases[40]) or any other standardised criteria, as assessed by scoring above a cut-off on a validated rating scale.

Our secondary outcomes are the frequency of depression, anxiety or PTSD at 1 to 6 months postintervention, and at 7 to 24 months postintervention; the symptoms of depression, anxiety and psychological distress at each timepoint; the changes in service utilisation and contact coverage (including admission rates to hospital whether related to mental disorder or not; attendance rates with regards to utilisation of primary or community services or increased demand and/or referral rates from the primary/community setting to a mental health specialist); the resource use (for health services: personnel time allocated/number of consultations, other opportunity costs of the intervention or other aspects of the health service, assets; for participants: extra costs of travel, productivity, employment status, income, work absenteeism, retention, educational attainment, presenteeism).

We will group outcomes into three time points: (a) postintervention (0 to 1 month after the intervention); (b) follow-up 1 (1 to 6 months postintervention); (c) follow-up 2 (7 to 24 months postintervention).

## Search methods for identification of studies

We will adopt the search strategy of a recent Cochrane review of promotion and prevention interventions delivered through task-shifting approaches among children, adolescents and adults in LMICs.[28] In short, the following electronic databases were inspected without language restrictions: Epistemonikos, Cochrane Controlled Trials Register (CENTRAL), MEDLINE (see online supplemental appendix 2), Embase, PsycINFO, CINAHL, Global Index Medicus, ClinicalTrials.gov (Ctgov), International Clinical Trials Registry Platform (ICTRP). To ensure the robustness of the search strategy, we will cross check the results with those of a Cochrane review focused on mental health preventative strategies in humanitarian settings.[29]

In addition to electronic search, we will perform a manual search of reference lists from relevant systematic reviews and primary studies related to this topic. We will document excluded and included studies following the PRISMA guidelines.[32]

## Data collection and analysis
### Selection of studies

All titles and abstracts identified through electronic searches will be downloaded into the reference management database (EndNote), and duplicates will be removed. The study selection process will be conducted independently by two reviewers (EP, BB or RS) using the software Rayyan.[41] Any discrepancies will be resolved through consensus or, if necessary, by the decision of a third reviewer (CL or MP).

### Data extraction and management

At least two reviewers (EP, BB, or RS) will extract descriptive characteristics and outcome data for the included studies using a standardised electronic spreadsheet. For all the included studies, the following information will be extracted: bibliographic information, setting, characteristics of participants, type of intervention (eg, nature of the intervention, modality/method of delivery, domain of social determinants, type of prevention), type of comparator, outcome measure(s), effect estimates.

Prior to data collection, we will pilot the form on at least four included studies. Disagreements will be resolved by discussion or involving a third author (CL or MP). We will contact authors of included articles to request additional data not included in the original publication.

## Data analysis
### Measures of treatment effect

We will conduct meta-analyses using the software Cochrane Review Manager V.5.4.[42] As we expect some degree of heterogeneity, we will first apply a random-effects model. A fixed-effects model will then be applied to check for any potential differences. We will estimate the effect of the intervention by using risk ratio, together with the 95% CI, for primary and secondary (non-economic) outcomes. For continuous data, we will use mean differences when the same rating scale is applied to measure the outcome of interest. We will calculate standardised mean differences (SMD) wherever different scales were used in RCTs to measure the same outcome of interest, together with 95% CI.[33] For economic data, we will present the results using a narrative approach.

If frequency of depression, anxiety or PTSD (primary outcome) are not available, we will first contact the authors of the studies. In the case of non-response, the frequency will be imputed according to commonly used cut-off scores for continuous measures of depressive, anxiety and PTSD symptoms from validated rating scales.[43] When SDs are not reported and not supplied by authors on request, we will calculate them based on other measures reported in the study, for example, SEs, t-statistics or p values, according to Altman.[44] For cluster-randomised controlled trails, we will include data adjusted with an intra-cluster correlation coefficient (ICC). If the ICC value is not reported or is not available from trial authors directly, we will assume it to be 0.05.[33]

If there is a sufficient number of included studies (n=10), publication bias will be examined by inspecting the funnel plot.

### Data synthesis

We will conduct separate meta-analysis for children and adolescents (≤19 years of age) and adults (>19 years of age). RCTs including mixed population groups will be allocated according to the proportion of participants belonging to the groups. In line with Cochrane methodology, we will not meta-analyse economic outcomes.

If there is a sufficient number of included studies, we will conduct the following subgroup analyses for the primary outcome: social determinants domain (demographic, economic, neighbourhood, environmental events and sociocultural), type of prevention (universal, selective, indicated), type of providers (PHWs, CWs), gender of the participants (male, female, others), country income at the time of study's implementation (low or lower middle or upper middle), humanitarian setting (vs non-humanitarian setting). We will carefully assess the type of control groups and the effects as a function of the control group (and type of control group).

We will assess heterogeneity by visual inspection of forest plots, and using the $I^2$ statistics, following the interpretation suggested by the Cochrane handbook[43]: 0%–40%: might not be important; 30%–60%: may represent moderate heterogeneity; 50%–90%: may represent substantial heterogeneity; 75%–100%: considerable heterogeneity.

We will perform sensitivity analyses for the primary outcome defined a priori to assess the robustness of our conclusions and to explore its impact on effect sizes. This will involve: excluding trials with a high risk of bias, excluding trials with methodological characteristics such

as variability in study design (cluster RCT vs individual RCT) or outcome measurement tools that might generate the highest heterogeneity in meta-analyses ($I^2 > 75\%$).

## Risk of bias and quality of included studies

Two independent reviewers (EP, BB) will apply the Cochrane risk-of-bias tool2 for RCTs (RoB V.2.0 tool).[33] This tool identifies key domains through which bias is likely to be included into trial design, conduct or analysis. For each of the five domains, the tool identifies potential source of bias, prompts the reviewer to provide support for a judgement and requests the review author's judgement concerning the level of bias (low, some concerns, high risk of bias). We will apply the RoB V.2.0 tool to the primary outcome. Any discrepancies will be resolved with a consensus between the two authors (EP and BB) and/or with a third reviewer (CL and MP). We will not exclude studies on the basis of their risk of bias. We will evaluate cluster-RCTs using specific section in the RoB V.2.0.[33] We will present our key findings in a summary of findings' table using the Grading of Recommendations, Assessment, Development and Evaluation (GRADE) approach.[45] We will produce GRADE tables for the primary outcome.

## Author affiliations

[1]WHO Collaborating Centre for Research and Training in Mental Health and Service Evaluation, Department of Neuroscience, Biomedicine and Movement Sciences, Section of Psychiatry, University of Verona, Verona, Italy
[2]Faculty of Biomedical Sciences, Institute of Public Health, Università della Svizzera Italiana, Lugano, Switzerland
[3]Transcultural Psychosocial Organization (TPO) Nepal, Kathmandu, Bagmati, Nepal
[4]Centre for Global Mental Health, Health Service and Population Research Department, Institute of Psychiatry, Psychology & Neuroscience, King's College London, London, UK
[5]Department of Psychiatry, University of Geneva, Geneve, Switzerland
[6]Innovations for Poverty Action, Bogotà, Colombia
[7]School of Government, Universidad de Los Andes, Bogota, Colombia
[8]Alan J Flisher Centre for Public Mental Health, Department of Psychiatry and Mental Health, University of Cape Town, Cape Town, South Africa
[9]Cochrane Global Mental Health, University of Verona, Verona, Veneto, Italy

**Contributors** CL, MJDJ, CB, MP, EP made substantial contributions to the conception and design of the study. The first draft of the manuscript was written by EP and critically revised by all authors (EP, BB, RS, EA, DT, MCDS, KS, NPL, ECG, MP, CB, MJDJ, CL) regarding their content's expertise. All authors read and approved the final manuscript.

**Funding** RS, DT, MCD, KS, NL, EG, MJ and CL are supported by the National Institute for Health Research (NIHR) (using the UK's Official Development Assistance (ODA) Funding) and Wellcome (grant number: 221940/Z/20/Z) under the Department of Health and Social Care (DHSC)-Wellcome Partnership for Global Health Research. The views expressed are those of the authors and not necessarily those of the Wellcome Trust, NIHR or the DHSC.

**Competing interests** None declared.

**Patient and public involvement** Patients and/or the public were not involved in the design, or conduct, or reporting, or dissemination plans of this research.

**Patient consent for publication** Not applicable.

**Provenance and peer review** Not commissioned; externally peer-reviewed.

**ORCID iDs**
Eleonora Prina http://orcid.org/0000-0002-4005-1967
Beatrice Bano http://orcid.org/0009-0001-9596-7622
Rakesh Singh http://orcid.org/0000-0002-0016-6903
Nagendra P Luitel http://orcid.org/0000-0002-8291-0205
Marianna Purgato http://orcid.org/0000-0002-3783-8195
Mark J D Jordans http://orcid.org/0000-0001-5925-8039
Crick Lund http://orcid.org/0000-0002-5159-8220

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
