## [Reviewer comments · BMJ Open]

ARTICLE DETAILS

TITLE (PROVISIONAL)	Effectiveness of combining prevention psychological interventions with interventions that address the social determinants of mental health in low- and middle- income countries: protocol of a systematic review and meta-analysis
AUTHORS	Prina, Eleonora; Bano, Beatrice; Singh, Rakesh; Albanese, Emiliano; Trujillo, Daniela; Dedios Sanguinetti, María Cecilia; Sorsdahl, Katherine; Luitel, Nagendra; Garman, Emily; purgato, marianna; Barbui, Corrado; Jordans, Mark J. D.; Lund, Crick

VERSION 1 – REVIEW

REVIEWER	Jennings, Hannah University of York, Department of Health Sciences
REVIEW RETURNED	22-Jan-2024

GENERAL COMMENTS	This is a well-written protocol of a systematic review and meta-analysis, that will report on some important findings. It is clear that it was well thought through and planned. I look forward to reading the findings! I have just a few minor comments and queries about the manuscript. They are: 1. Introduction: page 4, lines 17-20 reads “Accordingly, a Cochrane review focused on prevention interventions in humanitarian settings, found that psychological counselling may be effective for reducing depressive symptoms”. I do not think counselling is normally classed as “preventative”.2. Why are only interventions that include task-shifting included? An explanation/justification could be added.3. Data synthesis: how will it be decided whether there are sufficient number of studies for sub-group analysis?4. Page 9, line 32 – there is a typo (“).
--

REVIEWER	Tibber, Marc University College London, Department of Clinical, Educational and Health Psychology
REVIEW RETURNED	30-Jan-2024

GENERAL COMMENTS	The paper describes the protocol for a systematic review and meta-analysis of interventions that address psychological and social determinants of mental health in low- and middle- income countries. The protocol is clearly written, methodologically sound
---

	as far as I can tell, and addresses a question in an important area of research of practical real-world importance. I have only minor points to raise / address below.  • Page 4 line 3: Please define (briefly) “universal, selective, and/or indicated interventions”. • Page 4 line 8-9: “These interventions offer practical support, aim*ed* at enhancing coping strategies, hope, and resilience.” (Presumed typo). • Page 4 line 33-37: “This systematic review and meta-analysis aims to comprehensively evaluate the impact of interventions that include psychological ingredients and target the social determinants of mental health.” Whilst psychological interventions have been briefly defined, and relevant systematic reviews described, there is no explanation in the introduction of what is meant by interventions that target “the social determinants of mental health”, nor description of relevant systematic reviews (if relevant). I think this should be briefly addressed in the introduction. • Page 5 line 14-15: “we will exclude participants scoring above a validated cut-off on a symptom checklist at baseline or with a formal diagnosis of mental disorders at the time of recruitment”. I think it would be good to briefly justify why. • Page 6 line 7: “We will only include interventions delivered through task-shifting”. Again, I think that it would be useful to (briefly) justify this. Is it for reasons of scalability / cost effectiveness for example? • Page 6 line 25-29: “(i.e., the Diagnostic and Statistical Manual of Mental Disorders (DSM) [37], International Classification of Diseases (ICD) [38] or any other standardized criteria, as assessed by scoring above a cut-off on a validated rating scale.” Brackets need closing. • Page 8: I wonder if you might consider subgroup analyses looking at effects as a function of control group, since previous work has highlighted the importance of control group selection in determining outcomes / effectiveness.
--	--

VERSION 1 – AUTHOR RESPONSE

Comments from the Reviewer #1

Dr. Hannah Jennings, University of York, UCL Institute for Global Health

This is a well-written protocol of a systematic review and meta-analysis, that will report on some important findings. It is clear that it was well thought through and planned. I look forward to reading the findings!

R#0 response: We thank the Reviewer for this encouraging feedback.

I have just a few minor comments and queries about the manuscript. They are:

1. Introduction: page 4, lines 17-20 reads “Accordingly, a Cochrane review focused on prevention interventions in humanitarian settings, found that psychological counselling may be effective for reducing depressive symptoms”. I do not think counselling is normally classed as “preventative”.

R#1 response: We acknowledge the reviewer’s concern regarding the use of the term “counselling” in this context. However, although the authors described and labeled the intervention as ‘psychological

counselling', it was administered to enhance mental health outcomes in individuals without a formal mental health diagnosis. We would prefer to adhere to the authors' reporting. To clarify that this was in fact a prevention intervention, we have also modified the sentence slightly to read: "...found that psychological counselling may be effective for reducing depressive symptoms (MD -7.50, 95% CI - 9.19 to -5.81) and anxiety symptoms (MD -6.10, 95% CI - 7.57 to -4.63) among individuals without a formal mental health diagnosis".

2. Why are only interventions that include task-shifting included? An explanation/justification could be added.

R#2 response: We have decided to focus only on interventions delivered through task-shifting, as a significant portion of interventions in low- and middle-income countries are carried out by non-specialists. This is due to substantial deficits in specialist human resources and a related treatment and prevention gap. Consequently, we have opted to specify our research question more precisely by including this inclusion criteria. In this way, the results of the review will have important clinical but also policy implications. We would also like to clarify that in the screening process only a few studies were excluded because of the delivery modality (most trials delivered interventions using task-shifting). This choice is aligned with those of two Cochrane reviews on prevention and care interventions in LMICs (van Ginneken et al., 2021; Purgato et al., 2023). We have incorporated a brief explanation in the methods section (p. 6), as follow:

"We will only include interventions delivered through task-shifting, e.g., by non-specialist health workers (PHWs) and/or community workers (CWs) (i.e., interventions delivered by providers without a formal training in mental health and/or specialization with a redistribution or delegation of healthcare tasks within workforces and communities) due to scalability and cost-effectiveness reasons [21,35,36]."

3. Data synthesis: how will it be decided whether there are sufficient number of studies for sub-group analysis?

R#3 response: *Thank you for this comment. We will follow the Cochrane handbook and conduct subgroup analyses when at least 10 studies are available.*

4. Page 9, line 32 – there is a typo (").

R#4 response: We have addressed the typo.

Comments from the Reviewer #2

Dr. Marc Tibber, University College London

The paper describes the protocol for a systematic review and meta-analysis of interventions that address psychological and social determinants of mental health in low- and middle- income countries. The protocol is clearly written, methodologically sound as far as I can tell, and addresses a question in an important area of research of practical real-world importance. I have only minor points to raise / address below.

R#5 response: Thank you for this important feedback.

- Page 4 line 3: Please define (briefly) "universal, selective, and/or indicated interventions".

R#6 response: We have introduced a brief description of the three types of prevention strategies in the introduction section (p. 4), as suggested. The text reads as follows:

"Preventive strategies can range from universal, selective, and/or indicated interventions according to the Institute of Medicine (IOM) Framework [22,25]. Specifically, universal prevention includes strategies for the whole population, independently from the exposure to risk factors. Selective prevention targets high risk subpopulations based on identifiable risk factors (Lund et al., 2023) and indicated prevention focuses on individuals with increased vulnerability for a disorder (i.e., increased psychological symptoms) but not meeting the criteria for a full-blown mental health condition [22, 25]."

• Page 4 line 8-9: "These interventions offer practical support, aim*ed* at enhancing coping strategies, hope, and resilience." (Presumed typo).

R#7 response: We fixed the typo.

• Page 4 line 33-37: "This systematic review and meta-analysis aims to comprehensively evaluate the impact of interventions that include psychological ingredients and target the social determinants of mental health." Whilst psychological interventions have been briefly defined, and relevant systematic reviews described, there is no explanation in the introduction of what is meant by interventions that target "the social determinants of mental health", nor description of relevant systematic reviews (if relevant). I think this should be briefly addressed in the introduction.

R#8 response: Thank you for this comment. We have provided a clarification on the concept of 'intervention targeting social determinants' in the introduction (p. 4). A more comprehensive explanation has been added in the methods section, along with references on the theoretical framework. The text in the Introduction reads as follows:

"While there is preliminary evidence supporting the effectiveness of psychological interventions (Purgato et al., 2023) and programs targeting the social determinants of mental health (Oswald et al., 2024) in preventing mental health disorders, there is a notable absence of systematic reviews of trials combining social (i.e., seeking to change or improve a social determinant of mental health) and psychological components. Only through our review can we understand the efficacy of both ingredients (i.e., psychological, and social) and approach a 'component' meta-analysis."

• Page 5 line 14-15: "we will exclude participants scoring above a validated cut-off on a symptom checklist at baseline or with a formal diagnosis of mental disorders at the time of recruitment". I think it would be good to briefly justify why.

R#9 response: We will exclude participants with a score above a validated cut-off on a symptom checklist at baseline or with a formal diagnosis of mental disorder at enrollment, as our focus is on evaluating the effectiveness of prevention interventions. Therefore, we aim to include participants without a mental health condition (differently from treatment interventions). This criterion was also used in a recent Cochrane systematic review and meta-analysis on prevention interventions in low- and middle-income countries (Purgato et al., 2023), and in RCTs testing MHPSS interventions (Purgato et al., 2021; Acarturk et al., 2022; Turrini et al., 2022; Cuijpers, 2022). Additionally, we have specified the rationale ('given the focus on prevention of common mental health conditions') in the inclusion criteria (p. 5).

• Page 6 line 7: "We will only include interventions delivered through task-shifting". Again, I think that it would be useful to (briefly) justify this. Is it for reasons of scalability / cost effectiveness for example?

R#10 response: Thank you. See response 2, we have addressed the point.

• Page 6 line 25-29: "(i.e., the Diagnostic and Statistical Manual of Mental Disorders (DSM) [37], International Classification of Diseases (ICD) [38] or any other standardized criteria, as assessed by scoring above a cut-off on a validated rating scale." Brackets need closing.

R#11 response: We have fixed it.

• Page 8: I wonder if you might consider subgroup analyses looking at effects as a function of control group, since previous work has highlighted the importance of control group selection in determining outcomes / effectiveness.

R#12 response: Thank you for this suggestion. We will carefully assess the type of control groups and the effects as a function of the control group (and type of control group). We added a sentence in the analysis section to make this point.

Reviewer: 1

Competing interests of Reviewer: I have no competing interests.

Reviewer: 2

Competing interests of Reviewer: None.

VERSION 2 – REVIEW

REVIEWER	Jennings, Hannah University of York, Department of Health Sciences
REVIEW RETURNED	09-Apr-2024

GENERAL COMMENTS	I am satisfied that all my comments have been addressed. I am happy for this manuscript to proceed to publication.
--

REVIEWER	Tibber, Marc University College London, Department of Clinical, Educational and Health Psychology
REVIEW RETURNED	29-Apr-2024

GENERAL COMMENTS	I am happy that all my points have been adequately addressed. I look forward to reading this interesting review once it is published!
---

VERSION 2 – AUTHOR RESPONSE